

# Attention-enhanced gated recurrent unit for action recognition in tennis

Meng Gao[1] and Bingchun Ju[2]

[1] College of Sports and Health Management, Henan Finance University, Zhengzhou, China
[2] College of Sports, Zhengzhou University of Light Industry, Zhengzhou, China

## ABSTRACT

Human Action Recognition (HAR) is an essential topic in computer vision and artificial intelligence, focused on the automatic identification and categorization of human actions or activities from video sequences or sensor data. The goal of HAR is to teach machines to comprehend and interpret human movements, gestures, and behaviors, allowing for a wide range of applications in areas such as surveillance, healthcare, sports analysis, and human-computer interaction. HAR systems utilize a variety of techniques, including deep learning, motion analysis, and feature extraction, to capture and analyze the spatiotemporal characteristics of human actions. These systems have the capacity to distinguish between various actions, whether they are simple actions like walking and waving or more complex activities such as playing a musical instrument or performing sports maneuvers. HAR continues to be an active area of research and development, with the potential to enhance numerous real-world applications by providing machines with the ability to understand and respond to human actions effectively. In our study, we developed a HAR system to recognize actions in tennis using an attention-based gated recurrent unit (GRU), a prevalent recurrent neural network. The combination of GRU architecture and attention mechanism showed a significant improvement in prediction power compared to two other deep learning models. Our models were trained on the THETIS dataset, one of the standard medium-sized datasets for fine-grained tennis actions. The effectiveness of the proposed model was confirmed by three different types of image encoders: InceptionV3, DenseNet, and EfficientNetB5. The models developed with InceptionV3, DenseNet, and EfficientNetB5 achieved average ROC-AUC values of 0.97, 0.98, and 0.81, respectively. While, the models obtained average PR-AUC values of 0.84, 0.87, and 0.49 for InceptionV3, DenseNet, and EfficientNetB5 features, respectively. The experimental results confirmed the applicability of our proposed method in recognizing action in tennis and may be applied to other HAR problems.

## INTRODUCTION

Action recognition is one of the essential tasks in computer vision and artificial intelligence, aiming to understand and classify human actions from video or sequences of pictures (*Krüger et al., 2007*). This field has witnessed significant advancements in recent years, driven by the availability of large-scale annotated datasets, powerful deep learning methods, and enhanced computational resources (*Abu-Bakar, 2019*). Convolutional

Corresponding author
Meng Gao, gaomeng4343@163.com

neural networks (CNNs) (*Yao, Lei & Zhong, 2019*) and recurrent neural networks (RNNs) (*Richard & Gall, 2017*) have been widely employed to extract spatiotemporal features and model the temporal dependencies within action sequences. State-of-the-art methods, such as two-stream CNNs (*Zhu et al., 2019*; *Zhao et al., 2020*; *Xiong et al., 2020*) and 3D CNNs (*Ji et al., 2013*; *Yang et al., 2019*; *Ouyang et al., 2019*), have demonstrated remarkable performance in recognizing actions in diverse contexts, including sports analysis, surveillance (*Lin, Wang & Li, 2022*), and human–computer interaction (*Jannat et al., 2023*; *Lim et al., 2020*). However, there are challenges (*e.g.*, fine-grained action recognition, real-time processing, *etc.*) that continue to drive research in this dynamic field (*Jegham et al., 2020*; *Pareek & Thakkar, 2020*). Therefore, the exploration of novel and more efficient methods to further improve the recognition ability of intelligent systems is necessary.

Deep learning (DL) has emerged as a powerful technique for extracting discriminative and salient features in high-level action and behavior recognition from video data (*Dai et al., 2019*; *Zhang et al., 2021*; *Chen et al., 2022*). Present DL methods employed in human action recognition (HAR) are constructed based on basic CNNs for extracting features from video frames through the utilization of pre-trained models. These convolutional layers are responsible for capturing spatial features essential for model classification (*Lu et al., 2023*). However, traditional CNN models exhibit comparatively lower performance than manually crafted features when applied to sequential data (*Khemchandani & Sharma, 2016*). Notably, widely used CNN architectures like AlexNet (*Krizhevsky, Sutskever & Hinton, 2017*), VGGNet (*Simonyan & Zisserman, 2014*), and ResNet (*He et al., 2016*) primarily focus on extracting spatial features from individual input images, thus proving to be less effective in capturing temporal information critical for HAR within video sequences. *Dai, Liu & Lai (2020)* introduced long short-term memory (LSTM) networks incorporating spatiotemporal information learned through CNNs for action recognition. To address the challenge of capturing dynamic information in sequential data, *Kwon et al. (2018)* used advanced video-based HAR techniques with two-stream approaches characterized by two separate modules dedicated to learning spatial and temporal features. These modules were specially designed with mechanisms for fusion aimed at capturing the evolving information within video sequences. *Meng, Liu & Wang (2018)* have tackled spatiotemporal aspects by employing LSTM models explicitly designed for long-term video sequences to capture and process temporal features in the context of HAR in surveillance systems. Most recently, *Muhammad et al. (2021)* proposed using attention-based LSTM with dilated CNN features for action recognition. Accurately recognizing human actions in real-world videos remains a challenge due to the lack of crucial information about motion, style, and background clutter. Traditional methods struggle with continuous actions, crowded scenes, and noise issues (*Baccouche et al., 2011*). Similarly, current computational methods have improved sequence learning using RNNs, LSTM, and gated recurrent units (*Le et al., 2019*), but they often overlook important details within sequences, which are essential for connecting preceding and succeeding frames.

Sports analytics is the application of data analysis and statistical techniques to gain valuable insights and make informed decisions in the world of sports. It involves the

collection and analysis of various data sources, including player performance statistics, game metrics, and even fan engagement data. Sports analytics has revolutionized the way teams, coaches, and organizations approach player recruitment, game strategy, and injury prevention. By leveraging advanced analytics tools and techniques, sports professionals can optimize player performance, make data-driven decisions, and enhance the overall sports experience for both athletes and fans. It has become an integral part of modern sports, driving innovation and competitiveness in the industry. The field of sports analytics is experiencing significant growth, largely due to the explosion of accessible big data in this domain (*Morgulev, Azar & Lidor, 2018*; *Apostolou & Tjortjis, 2019*; *Sarlis & Tjortjis, 2020*). Traditionally, sports data was gathered manually and primarily comprised match outcomes and basic statistics, like the percentage of successful first serves in tennis. However, in recent times, the availability of spatiotemporal data, such as player positions and higher-level information, has expanded the possibilities for in-depth analysis (*Mora & Knottenbelt, 2017*). Under the scope of our study, we focus on tennis action recognition. Several published works were done on this topic with interesting findings. *Zhu et al. (2006)* introduced the HAR system to classify actions as 'left-swing' and 'right-swing' using support vector machines (SVM) and video descriptors based on optical flow. In another study, *FarajiDavar et al. (2011)* developed a classification system to distinguish tennis actions, including 'non-hit', 'hit', and 'serve'. Unfortunately, these experiments were conducted using the ACASVA Actions dataset (*De Campos et al., 2011*), which is not allowed for re-distribution. This dataset provides features and labels but lacks access to the RGB videos. Hence, to conduct our study, a publicly available dataset is more suitable for model development as well as comparisons with other relevant methods. Among the various datasets available for action recognition in tennis, THETIS (*Gourgari et al., 2013*) is a suitable dataset conveniently aligned with our objectives. In this study, we propose using an attention-based gated recurrent unit (GRU) architecture to perform action recognition in tennis. Our primary objective is to construct a model that is capable of effectively classifying these videos into the 12 fine-grained action categories.

# EXPERIMENTS

## Dataset

Introduced in 2013, the THree dimEnsional TennIs Shots (THETIS) dataset is one of the standard medium-sized datasets for fine-grained tennis actions, encompassing video clips featuring 55 different individuals executing 12 distinct tennis maneuvers multiple times (*Gourgari et al., 2013*). These videos are captured in the RGB format, characterized by low-definition quality, monocular perspective, and in-the-wild settings, featuring dynamic backgrounds and occasional occlusions. In the THETIS dataset, tennis shots were executed by a combination of 31 novice and 24 skilled players. Multiple repetitions of each shot were recorded, yielding a total of 8,734 videos (each cropped to a single period) that were subsequently converted into AVI format. These videos collectively span 7 h and 15 min. In our study, we used a subset of THETIS data consisting of 1,980 RGB videos of size $640 \times 480$. In each video, a player performs an action that corresponds to one of 12 possible tennis strokes.

| Table 1 | Information about the datasets used in the study. |
|---|---|
| **Data** | **Number of samples (videos)** |
| Training | 1,584 |
| Validation | 204 |
| Test | 192 |
| **Total** | **1,980** |

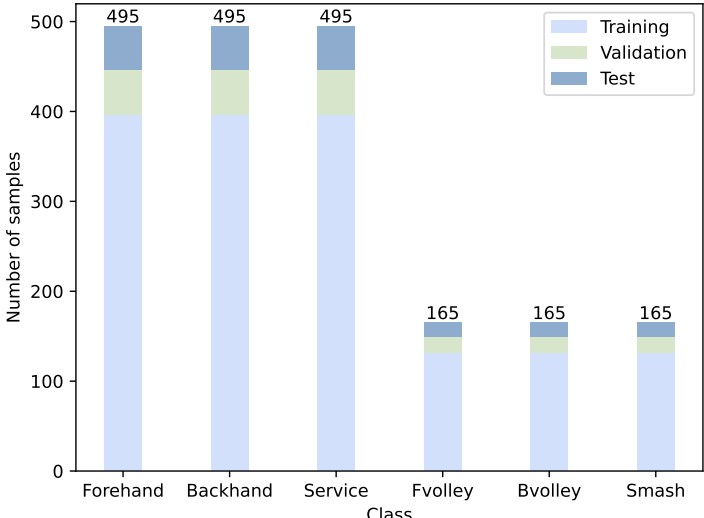

**Figure 1   Number of video samples for each class.**

## Data sampling

To simplify the classification task, we grouped 12 categories of actions (labels) into six new categories representing fundamental tennis strokes, including 'forehand', 'backhand', 'forehand volley', 'backhand volley', 'serve', and 'smash'. For each action, 16 cropped frames were obtained to represent the entire video. For each video, the cropped frames were carefully checked to ensure they were arranged in precise order to avoid failures in learning. We split all the videos into three datasets: a training set, a validation set, and a test set. Table 1 shows the number of video samples in each set, and Fig. 1 presents the distribution of number of videos corresponding to each class.

## Proposed model
### Model architecture

Figure 2 describes the model architecture proposed in our study. Each video is cropped into 16 frames for action recognition. These frames are then embedded using pre-trained encoders (*Szegedy et al., 2016*) to create corresponding vectors of size 15×2048. Three different pre-trained models, including InceptionV3 (*Szegedy et al., 2016*), DenseNet (*Huang et al., 2017*), and EfficientNetB5 (*Tan & Le, 2019*), were used for featurization. These vectors then enter two gated recurrent unit (GRU) layers of hidden dimension sizes

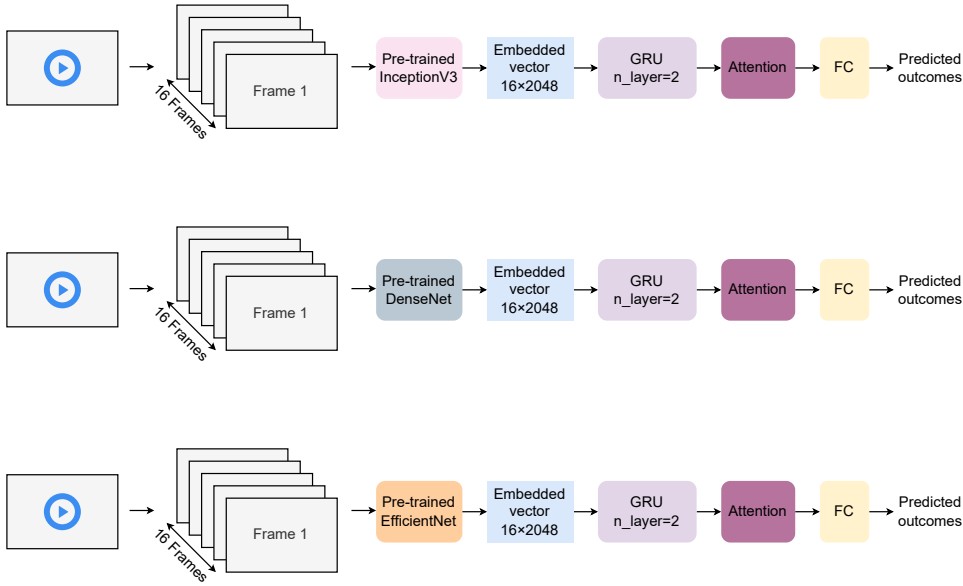

**Figure 2  Architecture of the proposed HAR model.**

of 256, respectively. The output features are learned in the attention layer to selectively learn essential characteristics. Finally, the attended features are transferred to the fully connected (FC) layer to return final outcomes. The model was trained over 100 epochs with a batch of 32 samples. The learning rate was fixed at 0.0001, and the training time per epoch is estimated to be about 0.2 s.

### Attention layer

An attention layer constitutes a pivotal component within DL architectures, prominently featured in natural language processing and computer vision domains (*Vaswani et al., 2017*). This layer facilitates the selective weighting of input data elements, enabling models to emphasize pertinent information while diminishing the significance of extraneous details. This mechanism closely mirrors human attention mechanisms, rendering it particularly efficacious in tasks such as machine translation, sentiment analysis, and image captioning. Adding attention layers to models helps them understand context better and capture long-range dependencies. This is a key part of getting better results in many AI tasks.

In this work, the attention layer was designed to detect which source objects are associated with the next target object and assign suitable attention weights when computing the context vector $c_j$. Given embedded source representation $H = \{h_1, h_2, h_3, \ldots, h_n\}$ and the previous decoded state $s_{j-1}$, $c_j$ can be simply expressed as:

$$c_j = Attention(H, s_{j-1}). \tag{1}$$

Initially, the attention weight $\alpha_{j,i}$ is computed to estimate the degree of association of a source object $x_i$ when predicting the target object $y_j$ *via* a feed-forward network:

$$\alpha_{j,i} = \frac{exp(e_{j,i})}{\sum_k exp(e_{j,k})}, \tag{2}$$

where the association score $e_{j,i}$ is derived from *Bahdanau, Cho & Bengio (2014)* as:

$$e_{j,i} = v_\alpha^T \odot tanh(W_\alpha \cdot s_{j-1} + U_\alpha \cdot h_i). \tag{3}$$

Basically, the greater attention weight $\alpha_{j,i}$ refers to the higher importance of object $x_i$ for the next object prediction. Hence, Attention computation creates $c_j$ by directly assigning weights for the source representations $H$ with their corresponding attention weights $\alpha_{i-1}^n$:

$$c_j = \sum_i \alpha_{j,i} \cdot h_i. \tag{4}$$

### Image embedding

Image embedding in DL refers to the process of transforming high-dimensional image data into a lower-dimensional representation that retains essential information about the image's content and characteristics. This lower-dimensional representation, known as an image embedding or feature vector, is designed to capture semantically meaningful features and patterns within the image. Image embeddings are valuable because they facilitate various computer vision tasks, such as image retrieval, object detection, and image similarity analysis. Deep learning models, especially CNNs, are commonly employed to generate image embeddings by extracting hierarchical and abstract features from the input image. The resulting image embeddings are not only more compact but also contain valuable semantic information, making them suitable for applications where efficient and meaningful image representations are required.

InceptionV3 is a pre-trained CNN model developed by Google (*Szegedy et al., 2016*). It is now known as one of the effective inception modules that facilitate multi-scale feature extraction. These modules enable the network to effectively recognize objects and patterns of different sizes within images. InceptionV3 also incorporates optimizations like batch normalization and factorized convolutional layers to enhance training stability and reduce computational complexity. Trained on large datasets like ImageNet (*Deng et al., 2009*), InceptionV3 is widely used in computer vision for tasks such as image recognition and object detection due to its efficiency and robust performance.

DenseNet is another pre-trained CNN model designed for computer vision tasks (*Huang et al., 2017*). Its unique feature is dense connectivity, where each layer directly connects to every other layer, enabling efficient feature reuse and better gradient flow. DenseNet models, like DenseNet-121 and DenseNet-169, have gained popularity for their ability to learn rich features effectively, making them a top choice for tasks like image classification, object detection, and segmentation due to their superior performance and parameter efficiency.

EfficientNet, a pioneering deep learning architecture developed by Google, has been known for its exceptional computational efficiency and concurrent model performance improvement (*Tan & Le, 2019*). This innovation relies on compound scaling, simultaneously optimizing network depth, width, and resolution, resulting in notable advancements in various computer vision tasks. Its capacity to strike a delicate balance between model complexity and predictive accuracy has made EfficientNet a preferred

choice for machine learning practitioners, offering quicker training and deployment while maintaining competitive state-of-the-art performance. This scalability and adaptability have profound implications for resource-efficient and versatile deep learning solutions across diverse applications in artificial intelligence.

## Metrics

We used a series of metrics to assess the performance of models, including Area under the ROC Curve (ROC-AUC), Area under the PR Curve (PR-AUC), Accuracy (ACC), Matthews Correlation Coefficient (MCC), F1 score (F1), Recall (REC), and Precision (PRE). The mathematical formula of these metrics are expressed as:

$$ACC = \frac{TP + TN}{TP + TN + FP + FN}, \tag{5}$$

$$MCC = \frac{TP \times TN - FP \times FN}{\sqrt{(TP + FP)(TP + FN)(TN + FP)(TN + FN)}}, \tag{6}$$

$$REC = \frac{TP}{TP + FN}, \tag{7}$$

$$PRE = \frac{TP}{TP + FP}, \tag{8}$$

$$F1 = 2 \times \frac{PRE \times REC}{PRE + REC}, \tag{9}$$

where TP, TN, FP, and FN are the numbers of True Positive, True Negative, False Positive, and False Negative samples.

# RESULTS AND DISCUSSION

## Model performance

Table 2 provides information on the performance of models on the test set across three types of features. To assess the performance of our proposed model, we implemented two conventional machine learning models, including Random Forest (RF) (*Breiman, 2001*) and SVM (*Cristianini & Ricci, 2008*), and two other deep learning models based on LSTM and CNN architectures. These two architectures are commonly employed to construct the HAR systems. In general, models developed using EfficientNetB5 work less effectively than those developed using InceptionV3 and DenseNet features. For InceptionV3, the LSTM and CNN models achieve equivalent performance with ROC-AUC and PR-AUC values of 0.93 and 0.70, respectively. Except for MCC, these two models' other metrics are equal. Our model obtains ROC-AUC and PR-AUC values of 0.94 and 0.75, which are higher than those of the LSTM and CNN models. Besides, in terms of other metrics, our model shows higher values. While the SVM model achieves an equivalent ROC-AUC value as ours, the

**Table 2  The performance of models on test set. Bold indicates the highest value corresponding to a specific pair of feature type and evaluation metric.**

| Feature type | Model | ROC-AUC | PR-AUC | ACC | MCC | F1 | REC | PRE |
|---|---|---|---|---|---|---|---|---|
| | RF | 0.88 | 0.60 | 0.66 | 0.57 | 0.42 | 0.46 | 0.46 |
| | SVM | **0.94** | 0.72 | 0.69 | 0.61 | 0.48 | 0.51 | 0.49 |
| Inceptionv3 | CNN | 0.93 | 0.70 | 0.76 | 0.70 | 0.65 | 0.64 | 0.70 |
| | LSTM | 0.93 | 0.70 | 0.76 | 0.69 | 0.65 | 0.64 | 0.70 |
| | Ours | **0.94** | **0.75** | **0.77** | **0.71** | **0.66** | **0.67** | **0.72** |
| | RF | 0.88 | 0.61 | 0.65 | 0.55 | 0.39 | 0.44 | 0.41 |
| | SVM | 0.94 | 0.73 | 0.63 | 0.52 | 0.36 | 0.42 | 0.32 |
| DenseNet | CNN | 0.95 | 0.74 | 0.80 | 0.74 | 0.72 | 0.70 | 0.77 |
| | LSTM | 0.96 | 0.80 | 0.79 | 0.73 | 0.73 | 0.73 | 0.73 |
| | Ours | **0.97** | **0.82** | **0.82** | **0.78** | **0.76** | **0.74** | **0.79** |
| | RF | **0.82** | **0.54** | **0.61** | **0.50** | **0.48** | **0.48** | **0.62** |
| | SVM | 0.75 | 0.40 | 0.34 | 0.13 | 0.19 | 0.23 | 0.18 |
| EfficientNetB5 | CNN | 0.80 | 0.45 | 0.47 | 0.32 | 0.39 | 0.40 | 0.40 |
| | LSTM | 0.79 | 0.46 | 0.48 | 0.32 | 0.36 | 0.37 | 0.51 |
| | Ours | **0.82** | 0.49 | 0.55 | 0.42 | 0.47 | 0.45 | 0.55 |

RF model shows limited predictive power. For other metrics, the SVM model is not as good as our proposed model. For DenseNet features, our model still outperforms the other two models with ROC-AUC and PR-AUC values of 0.97 and 0.82, respectively. The LSTM model trained with DenseNet features works better than the corresponding CNN model. The results demonstrate that the attention-based GRU (our proposed model) works more efficiently compared to these other deep learning models for recognizing actions in tennis. The performance of the SVM and RF models trained with this feature is not significant. For EfficientNetB5, our model and the RF model both achieve an equivalent ROC-AUC value of 0.82. However, the RF model achieves all recorded metrics with higher values compared to the others, including ours.

InceptionV3 (*Szegedy et al., 2016*), DenseNet (*Huang et al., 2017*), and EfficientNet (*Tan & Le, 2019*) are prominent pre-trained models based on CNN architectures in the field of computer vision, yet they exhibit distinct characteristics. The key similarity between them lies in their effectiveness for image classification tasks. Both networks have demonstrated remarkable performance in various image-related challenges, benefiting from their deep architectures and pre-trained model variants. However, a notable difference is in their architectural design. DenseNet employs densely connected blocks, enabling direct connections between all layers within a block and fostering feature reuse and gradient flow. In contrast, InceptionV3 relies on innovative inception modules, incorporating multiple filter sizes to capture features at different scales efficiently. Additionally, DenseNet tends to be more parameter-efficient due to its dense connectivity, while InceptionV3 emphasizes computational efficiency with factorized convolutional layers. The choice between these architectures often depends on the specific requirements of the task at hand, as both DenseNet and InceptionV3 offer strong capabilities but excel in different aspects. Compared to InceptionV3 and DenseNet, EfficientNet stands out for its parameters

**Table 3  The performance of our models on the test set over multiple trials.**

| Feature type | Trial | ROC-AUC | PR-AUC | ACC | MCC | F1 | REC | PRE |
|---|---|---|---|---|---|---|---|---|
| | 1 | 0.96 | 0.83 | 0.83 | 0.79 | 0.77 | 0.76 | 0.80 |
| | 2 | 0.97 | 0.88 | 0.85 | 0.82 | 0.82 | 0.81 | 0.83 |
| | 3 | 0.98 | 0.86 | 0.88 | 0.84 | 0.82 | 0.81 | 0.84 |
| | 4 | 0.96 | 0.81 | 0.83 | 0.79 | 0.78 | 0.79 | 0.80 |
| | 5 | 0.97 | 0.83 | 0.81 | 0.77 | 0.79 | 0.82 | 0.77 |
| | 6 | 0.98 | 0.89 | 0.85 | 0.81 | 0.79 | 0.79 | 0.81 |
| Inceptionv3 | 7 | 0.96 | 0.79 | 0.82 | 0.77 | 0.77 | 0.78 | 0.76 |
| | 8 | 0.98 | 0.87 | 0.87 | 0.84 | 0.81 | 0.82 | 0.83 |
| | 9 | 0.97 | 0.83 | 0.85 | 0.81 | 0.79 | 0.76 | 0.86 |
| | 10 | 0.97 | 0.84 | 0.88 | 0.84 | 0.82 | 0.79 | 0.90 |
| | Mean | 0.97 | 0.84 | 0.85 | 0.81 | 0.80 | 0.79 | 0.82 |
| | SD | 0.01 | 0.03 | 0.02 | 0.03 | 0.02 | 0.02 | 0.04 |
| | 95%CI | (0.97, 0.97) | (0.82, 0.86) | (0.84, 0.86) | (0.79, 0.83) | (0.79, 0.81) | (0.78, 0.80) | (0.80, 0.84) |
| | 1 | 0.97 | 0.87 | 0.86 | 0.83 | 0.82 | 0.81 | 0.86 |
| | 2 | 0.97 | 0.87 | 0.87 | 0.84 | 0.84 | 0.81 | 0.92 |
| | 3 | 0.97 | 0.85 | 0.85 | 0.82 | 0.80 | 0.79 | 0.83 |
| | 4 | 0.97 | 0.83 | 0.80 | 0.76 | 0.76 | 0.76 | 0.77 |
| | 5 | 0.98 | 0.88 | 0.89 | 0.86 | 0.85 | 0.84 | 0.87 |
| | 6 | 0.98 | 0.91 | 0.90 | 0.87 | 0.86 | 0.86 | 0.87 |
| DenseNet | 7 | 0.98 | 0.87 | 0.89 | 0.85 | 0.86 | 0.87 | 0.86 |
| | 8 | 0.98 | 0.88 | 0.89 | 0.86 | 0.86 | 0.87 | 0.87 |
| | 9 | 0.97 | 0.86 | 0.89 | 0.85 | 0.86 | 0.85 | 0.86 |
| | 10 | 0.98 | 0.88 | 0.89 | 0.86 | 0.85 | 0.86 | 0.84 |
| | Mean | 0.98 | 0.87 | 0.87 | 0.84 | 0.84 | 0.83 | 0.85 |
| | SD | 0.01 | 0.02 | 0.03 | 0.03 | 0.03 | 0.03 | 0.03 |
| | 95%CI | (0.98, 0.98) | (0.86, 0.88) | (0.85, 0.89) | (0.82, 0.86) | (0.82, 0.86) | (0.81, 0.85) | (0.83, 0.87) |
| | 1 | 0.85 | 0.56 | 0.57 | 0.45 | 0.49 | 0.49 | 0.53 |
| | 2 | 0.79 | 0.46 | 0.51 | 0.38 | 0.43 | 0.43 | 0.50 |
| | 3 | 0.83 | 0.49 | 0.55 | 0.43 | 0.43 | 0.44 | 0.44 |
| | 4 | 0.87 | 0.55 | 0.61 | 0.51 | 0.52 | 0.51 | 0.60 |
| | 5 | 0.83 | 0.48 | 0.55 | 0.42 | 0.40 | 0.41 | 0.45 |
| | 6 | 0.84 | 0.54 | 0.51 | 0.38 | 0.45 | 0.45 | 0.50 |
| EfficientNetB5 | 7 | 0.76 | 0.42 | 0.54 | 0.40 | 0.42 | 0.42 | 0.55 |
| | 8 | 0.76 | 0.44 | 0.57 | 0.43 | 0.38 | 0.39 | 0.45 |
| | 9 | 0.79 | 0.47 | 0.52 | 0.37 | 0.41 | 0.41 | 0.49 |
| | 10 | 0.79 | 0.47 | 0.56 | 0.43 | 0.45 | 0.45 | 0.45 |
| | Mean | 0.81 | 0.49 | 0.55 | 0.42 | 0.44 | 0.44 | 0.50 |
| | SD | 0.04 | 0.04 | 0.03 | 0.04 | 0.04 | 0.04 | 0.05 |
| | 95%CI | (0.79, 0.83) | (0.46, 0.52) | (0.53, 0.57) | (0.39, 0.45) | (0.42, 0.46) | (0.42, 0.46) | (0.47, 0.53) |

and computational efficiency, making it an excellent choice for resource-constrained environments. However, the choice between these models should still consider the specific

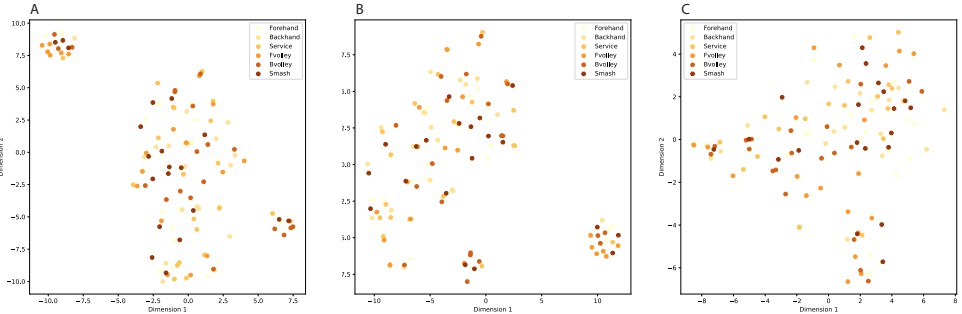

**Figure 3** **Class clustering across different feature types.** (A) InceptionV3, (B) DenseNet, (C) Efficient-NetB5.

needs of the task at hand, as each has its own strengths and trade-offs. Figure 3 visualizes class clustering across these three feature types.

## Performance variation assessment

Table 3 summarizes the variation in performance of models developed with InceptionV3, DenseNet, and EfficientNetB5 features. For the model developed with InceptionV3 features, the average ROC-AUC and PR-AUC values are 0.97 and 0.84, respectively. The standard deviations of ten trials are also small. Other metrics are higher than 0.8, with standard deviations of about 0.02. In terms of the model developed with DenseNet features, the average ROC-AUC and PR-AUC values are 0.98 and 0.87, respectively. Models developed with EfficientNetB5 obtained the lowest values for all metrics. Besides, for other metrics, their measured values for the models developed with DenseNet are higher than those recorded in those developed with InceptionV3 and EfficientNetB5. The standard deviations of metrics recorded on the EfficientNetB5 are the highest, followed by DenseNet's and InceptionV3's.

## CONCLUSIONS

HAR systems employ computer vision and machine learning techniques to automatically detect, classify, and analyze human actions and gestures in video sequences or sensor data. In our study, we developed a HAR system for recognizing actions in tennis. The results indicated that our proposed model is an efficient framework that can perform recognition tasks. Compared to other deep learning methods, the attention-based GRU model showed better performance. Briefly, although our model showed higher performance in both conditions, DenseNet features are more suitable than InceptionV3 and EfficientNetB5 features for building our HAR system in our study. The effectiveness of attention-based GRU demonstrated its application in addressing other action recognition tasks in the future.

### Funding

The authors received no funding for this work.

### Competing Interests

The authors declare there are no competing interests.

### Author Contributions

- Meng Gao conceived and designed the experiments, performed the experiments, analyzed the data, performed the computation work, prepared figures and/or tables, authored or reviewed drafts of the article, and approved the final draft.
- Bingchun Ju conceived and designed the experiments, analyzed the data, prepared figures and/or tables, authored or reviewed drafts of the article, and approved the final draft.

### Data Availability

The code and data used in the experiments are available in the Supplemental File. The data is from *Gourgari et al. (2013)* and it is available at THETIS: http://thetis.image.ece.ntua.gr.

### Supplemental Information

Supplemental information for this article can be found online at http://dx.doi.org/10.7717/peerj-cs.1804#supplemental-information.

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
