# Peer review of "Attention-enhanced gated recurrent unit for action recognition in tennis"

_PeerJ Computer Science, doi:10.7717/peerj-cs.1804_

## Round 0.1 · original submission · Major Revisions

Please find attached the reviews of your manuscript. The two reviewers have identified several issues with the current version, but we are open to considering a revised manuscript addressing their concerns.

Furthermore, it is worth noting that both reviewers strongly recommend performing more experiments, especially to compare your proposed models with other machine learning models.

**Language Note:** PeerJ staff have identified that the English language needs to be improved. When you prepare your next revision, please either (i) have a colleague who is proficient in English and familiar with the subject matter review your manuscript, or (ii) contact a professional editing service to review your manuscript. PeerJ can provide language editing services - you can contact us at copyediting@peerj.com for pricing (be sure to provide your manuscript number and title). – PeerJ Staff

Reviewer 1 ·

Basic reporting

The manuscript is prepared with professional English and sufficient background. The tables and figures are qualified for publication. However, there are major points that need to be addressed before being considered for publication.

Experimental design

(1) Authors please check the source code and Figure 2. Compared to the source code, Figure 2 seems to incorrectly visualize the model architecture. It is not "Two independent layers of GRU". According to your code provided, it should be "One GRU with n_layer=2".
(2) More experiments need to be done to conclude the performance of the proposed method. Before proposing this method, did authors tried to use more simple approaches, such as conventional machine learning algorithms.
(3) In "Performance Variation Assessment" section, the authors said that they performed the experiments five times. Repeating experiments five times is not enough to draw any conclusion about the model's robustness. Hence, I suggest authors perform the experiments with an additional number of trials and then calculate the 95%CI. A histogram is required to be included.

Validity of the findings

The author repeated their experiments a number of times but it's not enough to conclude about the effectiveness of the method proposed. Although authors computed some statistical metrics to demonstrate their method is sound, these metrics need to be recomputed based on a larger number of repeated experiments. The conclusion is well-stated.

Additional comments

Minor point:
- Box for "Pretrained DenseNet" has a boundary. Please remove it or create boundaries for all other boxes.

Cite this review as

Reviewer 2 ·

Basic reporting

- The authors have employed language in the manuscript that is both comprehensible and reasonably specific. The majority of the technical content is easily understood.
- The figures included in the manuscript provide good visual support. Readers will be able to distinguish the important details because the figures are in vector format. The descriptions and labels of each figure are written in an understandable and comprehensive manner.
- The authors have provided nearly all of the results that are relevant to their hypothesis in their presentation. This indicates that the subject has been discussed in great detail. Nevertheless, the quality of the manuscript will be elevated if the authors add additional specifics, as will be discussed in the "Additional comments" that follow.

Experimental design

- The study is consistent with the goals and scope of the journal.
- The methodology for the experiment is adequate and technically sound. On the other hand, as will be pointed out below, additional information should be included.
- The authors provided both the code and the data. Those who want to expand on this research or replicate the reported results will find this useful.

Validity of the findings

- In my opinion, the results of this study have practical applications.
- The conclusions that are given in the manuscript are well-written and relate to the main research question.

Additional comments

- The Introduction part can be extended a bit more.
- The manuscript makes some good progress in using pre-trained models like InceptionV3, but it lacks a citation to DenseNet that is needed for Lines 110–111.
- Adding another type of feature would make the models more reliable, which would boost the credibility of the study.
- The manuscript would also be better with visualizations that show the features that were used. This would make the models easier to understand.
- Even though attention mechanisms are used, the manuscript does not go into much detail about them. The reader would have a much better understanding of the model architecture if the details of this block were made clearer in the manuscript. Incorporating a full explanation of the attention mechanisms used would improve the technical depth of the manuscript.
- It is also more convincing if the authors include other machine learning models besides the two deep learning models.
- Line 152: It is unnecessary to provide the complete names of LSTM and CNN, as they have already been mentioned at the start of the manuscript.
- Need to include a comma between the equations.
- References: Some words require capitalization, such as "5G" and "3D".

Cite this review as

---

## Round 0.2 · accepted · Accept

The authors have addressed all of the reviewers' comments, and the reviewers are both happy with the revised manuscript. The manuscript is now ready for publication.

Reviewer 1 ·

Basic reporting

The revised version is significantly improved with an extended introduction, professional English, and sufficient background.

Experimental design

The authors conducted additional experiments with Machine Learning and Deep Learning architectures. The results indicated that their proposed method can work effectively.

Validity of the findings

Additional experimental repetition was performed to add more evidence to the statistical approach. The statistical evidence pointed out the proposed method is robust and stable.

Cite this review as

Reviewer 2 ·

Basic reporting

The revision meets all the standard requirements.

Experimental design

The experimental design is OK. The authors have conducted more experiments to show that their proposed framework is better than other baseline models. Source code and data are provided.

Validity of the findings

The findings are valid. I have no more comments for the revised manuscript.

Additional comments

All of my concerns have been addressed, and my suggestions have been incorporated into the revision. The manuscript is now ready to be considered for publication.

Cite this review as